Development of metastasis and survival prediction model of luminal and non-luminal breast cancer with weakly supervised learning based on pathomics

Liu Hui
Ying Linlin
Song Xing
Xiang Xueping
Wei Shumei 2307001@zju.edu.cn
Departments of Clinical Pathology, The Second Affiliated Hospital of Medical College of Zhejiang University , Hangzhou, Zhejiang , China
Wang Jincheng
Electronic publication date: 2025 Jan 21
Publication date: 2025
Volume: 13
Electronic Location ID: e18780
Received 2024 Aug 9; Accepted 2024 Dec 9
Copyright: © 2025 Liu et al.
Copyright year: 2025
Copyright holder: Liu et al.
License: This is an open access article distributed under the terms of the Creative Commons Attribution License, which permits unrestricted use, distribution, reproduction and adaptation in any medium and for any purpose provided that it is properly attributed. For attribution, the original author(s), title, publication source (PeerJ) and either DOI or URL of the article must be cited.
License URL: https://creativecommons.org/licenses/by/4.0/

Keywords: Breast cancer, Pathomics, Weakly supervised learning, Metastasis prediction model, Survival prediction model

Funding: National Natural Science Foundation of China 81602465 The work was supported by National Natural Science Foundation of China (No. 81602465). The funders had no role in study design, data collection and analysis, decision to publish, or preparation of the manuscript.

==============================
Objective

Breast cancer stands as the most prevalent form of cancer among women globally. This heterogeneous disease exhibits varying clinical behaviors. The stratification of breast cancer patients into risk groups, determined by their metastasis and survival outcomes, is pivotal for tailoring personalized treatments and therapeutic interventions. The pathological sections of radical specimens encompass a diverse range of histological information pertinent to the metastasis and survival of patients. In this study, our objective is to develop a deep learning model utilizing pathological images to predict the metastasis and survival outcomes for breast cancer patients.

Methods

This study utilized pathological sections from 204 radical mastectomy specimens obtained between January 2013 and December 2014 at the Second Affiliated Hospital of the Medical College of Zhejiang University. The 204 pathological slices were scanned and transformed into whole slide imaging (WSI), with manual labeling of all tumor areas. The WSI was then partitioned into smaller tiles measuring 512 × 512 pixels. Three networks, namely Densely Connected Convolutional Network 121 (DenseNet121), Residual Network (ResNet50), and Inception_v3, were assessed. Subsequently, we combined patch-level predictions, probability histograms, and Term Frequency-Inverse Document Frequency (TF-IDF) features to create comprehensive participants representations. These features served as the foundational input for developing a machine learning algorithm for metastasis analysis and a Cox regression model for survival analysis.

Result

Our results show that the Inception_v3 model shows a particularly robust patch recognition ability for estrogen receptor (ER) recognition. Our pathological model shows high accuracy in predicting tumor regions. The train area under the curve (AUC) of the Inception_v3 model based on supervised learning is 0.975, which is higher than the model established by weakly supervised learning. But the AUC of the metastasis prediction in training and testing sets is higher than value based on supervised learning. Furthermore, the C-index of the survival prediction model is 0.710 in the testing sets, which is also better than the value by supervised learning.

Conclusion

Our study demonstrates the significant potential of deep learning models in predicting breast cancer metastasis and prognosis, with the pathomic model showing high accuracy in identifying tumor areas and ER status. The integration of clinical features and pathomics signature into a nomogram further provides a valuable tool for clinicians to make individualized treatment decisions.

Introduction

Breast cancer, a prevalent malignancy affecting women globally, poses a substantial burden with its high incidence and mortality rates (Curigliano et al., 2019). This disease exhibits significant biological heterogeneity, characterized by various molecular subtypes that dictate diverse responses to treatments and prognoses (Curigliano et al., 2019). The 2011 St. Gallen International Breast Cancer Conference established a consensus among experts regarding breast cancer classification based on immunohistochemical detection of key biomarkers such as estrogen receptor (ER), progesterone receptor (PR), human epidermal growth factor 2 (HER2), and Ki-67 (Goldhirsch et al., 2011). Histopathology tissue analysis stands as the cornerstone in cancer diagnosis and prognosis, offering invaluable insights into disease characterization (Khened et al., 2021). The widespread adoption of immunohistochemical (IHC) staining for breast cancer subtyping has significantly advanced diagnostic precision. However, the utility of IHC is hampered by its inherent limitations, including time-consuming procedures and high costs.

The application of deep learning methods is an emerging technique for the automated extraction of high-throughput phenotypes from medical images (Cifci, Foersch & Kather, 2022). Recent studies have demonstrated that deep learning architectures can help predict patient outcomes from digital pathology images (Courtiol et al., 2019; Saillard et al., 2020; Liao et al., 2020; Reichling et al., 2020; Kleppe et al., 2018; Kang et al., 2022). A few studies have compared the performance of deep learning systems applied to biopsy samples or virtual biopsies and found a decline of predictive performance (Cifci, Foersch & Kather, 2022; Echle et al., 2022; Muti et al., 2021). Routine histopathology images of tumor tissue specimens stained with hematoxylin and eosin (H&E) encompass a wealth of valuable information, including tumor differentiation, tumor budding, lymphovascular invasion, and perineural invasion, among other factors (Cifci, Foersch & Kather, 2022). Notably, the morphological characteristics of breast cancer captured in H&E staining may reflect underlying molecular or genetic information (Wang et al., 2019; Liu et al., 2022). Digital pathology facilitates quantitative analysis through the examination of digitized whole slide imaging (WSI) of histological specimens using machine learning. Previous studies have demonstrated that machine learning approaches extracting morphological features from H&E stained WSI can predict the expression of several single biomarkers, such as ER and HER2, in breast cancer (Bychkov et al., 2021; Naik et al., 2020).

Breast cancer is generally classified into luminal and non-luminal types based on ER status. Compared to luminal types, non-luminal breast cancers tend to be more aggressive and are associated with a poorer prognosis (Dunnwald, Rossing & Li, 2007). Studies have been conducted based on deep learning to predict the classification of luminal and non-luminal breast cancer through radiomics and ultrasoundomics (Huang et al., 2023). However, there are a scarcity of models predicting the metastasis and prognosis of breast cancer with different molecular typing using machine learning through pathological sections of radical specimens. In this study, our aim was to develop and validate a deep learning pathomic model to predict the metastasis and prognosis of breast cancer. Establishing metastasis and prognosis models of breast cancer with different ER status not only saves the time and cost of immunohistochemical diagnosis, but also avoids the subjectivity of pathological diagnosis. At the same time, it also provides a valuable tool for clinicians to make individualized treatment decisions.

Materials and Methods

Patients and follow-up

In this study, we selected 204 participants diagnosed with breast cancer with molecular typing at the Department of Pathology, Second Affiliated Hospital of Zhejiang University School of Medicine, spanning from January 1st, 2013, to December 31st, 2014. As of August 2023, these encompass all radical breast cancer specimens, including metastasis information (153 cases) or survival information (141 cases). Ethical approval for this study was obtained from the Human Research Ethics Committee of the Second Affiliated Hospital of Zhejiang University School of Medicine (Approval number 2023 Lun Shen Yan No. (1035)). The corresponding H&E slices of the 204 participants were scanned into full electronic images in KFB format using the Ningbo Jiangfeng scanner (KF-PRO-400) under a 20X objective lens. Subsequently, they were converted into WSI format with a format converter, and all images were manually marked.

Clinical and pathology procedures

In our dataset, the clinical information includes the patient’s age, gender, menopausal status, and whether there was new adjuvant therapy. Histopathological data comprise the type of radical surgery specimen, histological type and grade, tumor size, vascular invasion, margin, immunohistochemical results of ER, PR, HER2, and Ki-67 based on the entire tumor surgical specimen, along with the molecular results of HER2. Based on the IHC results of ER status in breast cancer radical surgery specimens, all breast tumors were categorized into luminal and non-luminal subtypes. ER positivity was defined as ≥1% of tumor cells stained with ER. In the presence of ER positivity, all tumors were identified as luminal subtypes; in the absence of ER positivity, all tumors were identified as non-luminal subtypes. Each corresponding WSI image received a label consistent with the tumor classification.

In our collection of breast cancer radical specimens, there are 132 cases of luminal subtypes, among which 102 cases have metastasis information and 100 cases have survival information. Additionally, there are 72 cases of non-luminal subtypes, with 53 cases having metastasis information and 55 cases having survival information (Table 1). Unidentified cases were those for which we were unable to contact patients or their families, and therefore, we were not able to obtain metastasis or survival information. Figure 1 shows the workflow in developing the Metastasis and Survival Prediction Model of luminal and non-luminal breast cancer with deep learning based on pathological sections.

Table 1 Molecular typing and metastasis and survival information of 204 breast cancer participants.

	Number	Gender	Age	Ki-67	Metastasis	Survival	
Luminal (ER+)	132	Female (131)	33–89	1–90%	Non-metastasis (85)	Live (90)	
Male (1)	Metastasis (17)	Died (10)	
Unidentified (30)	Unidentified (32)	
Non-luminal (ER−)	72	Female (72)	27–80	1–90%	Non-metastasis (45)	Live (47)	
Metastasis (8)	Died (8)	
Unidentified (19)	Unidentified (17)	

Figure 1 Workflow in developing metastasis and survival prediction model of luminal and non-luminal breast cancer with deep learning based on pathomics.

WSI annotation

In addressing the challenge posed by large digitized images, we implemented a preprocessing strategy. Initially, we divided the WSI into smaller tiles measuring 512 × 512 pixels. This partitioning, executed at a resolution of 0.5 μm/pixel, followed a non-overlapping approach. Our primary goal was to ensure the integrity of the data, and to achieve this, we excluded all patch regions with exclusively white backgrounds. Additionally, we applied color normalization to the small tiles using the Reinhard method. Throughout the model training process, we exclusively utilized annotated regions of interest (ROI), with other regions remaining unused for training purposes.

Deep learning training

Considering the substantial dimensions and variability of the tumor images, our deep learning pipeline comprised two tiers of predictions: patch-level prediction and multi-instance learning-based WSI feature fusion. During the training process, we utilized the patient’s ER status as the training label for the corresponding patches. All patches associated with a given patient shared that patient’s ER status.

We employed Z-score normalization on the RGB channels to ensure a standardized normal distribution of image intensities, serving as the input for our model. Throughout the training phase, we integrated online data augmentations such as random cropping and random horizontal and vertical flipping (Ding et al., 2024). However, for the test patches, only normalization was applied.

For patch-level prediction, we evaluated three widely recognized convolutional neural networks: DenseNet121, ResNet50, and Inception_v3. These networks have demonstrated exceptional performance in the ImageNet classification competition.

To broaden the model’s applicability across cohorts characterized by substantial heterogeneity, we utilized transfer learning. This process involved initializing the model parameters with pre-trained weights from the ImageNet dataset. To improve generalization, we carefully configured the learning rate. In this study, we employed the cosine decay learning rate algorithm, presented as follows:

ηt=ηmini+12(ηmaxi−ηmini)(1+cos(TcurTiπ)).

In this notation, ηmini=0 denotes the minimum learning rate, while ηmaxi = 0.01 represents the maximum learning rate, and Ti=50 signifies the number of iteration epochs. Additional hyperparameter configurations are as follows: optimizer–stochastic gradient descent (SGD), and loss function—softmax cross entropy.

WSI feature fusion

After completing the training of our deep learning model, we proceeded to predict labels and their corresponding probabilities for all patches. These patch likelihoods were then consolidated using a classifier to derive the features at the WSI level. To aggregate the patch likelihoods, we devised two distinct methods: (1) Patch likelihood histogram (PLH) pipeline: This approach involved utilizing a histogram to depict the distribution of patch likelihoods within the WSI. This histogram effectively encapsulated the spread of likelihoods, offering a representation of the WSI. (2) Bag of words (BoW) pipeline: Inspired by both the histogram-based and vocabulary-based approaches, this pipeline utilized a Term Frequency-Inverse Document Frequency (TF-IDF) mapping for each patch. This resulted in the generation of a TF-IDF feature vector that represented the WSI. By implementing these two independent pipelines, we consolidated the initial patch-level prediction results, generating WSI-level features. These features can be utilized to provide characterization information for downstream metastasis analysis and survival analysis tasks.

Signature building

In this study, we combined patch-level predictions, probability histograms, and TF-IDF features to create comprehensive patient representations. These features served as the foundational input for developing a machine learning algorithm for metastasis analysis and a Cox regression model for survival analysis.

Furthermore, to enhance clinical applicability and assess the incremental prognostic value of the pathology signature, we applied linear Cox proportional hazards models with L2 regularization (c = 0.3) and no L1 regularization for all multimodal and unimodal models. We utilized Kaplan–Meier (KM) analysis to determine whether each model stratified patients into clinically significant groups. To define group membership, we evaluated percentile thresholds and selected the value that maximized the significance of the separation in the training set using the log-rank test. This process was conducted individually for overall survival (OS), where applicable. All p-values for KM analysis were computed using the multivariate log-rank test.

Statistical analysis

We applied the Shapiro-Wilk test to conduct statistical tests for assessing the normality of clinical features. Subsequently, we used the t-tests to analyze the significance of these clinical features. The specific analytical tools employed were as follows: Python code written in Python version 3.7.12. The Python packages used in the analysis included Pandas version 1.2.4, NumPy version 1.20.2, PyTorch version 1.8.0, Onekey version 3.1.3, OpenSlide version 1.2.0, SciPy version 1.7.3, scikit-learn version 1.0.2, and Lifelines version 0.27.0.

Results

This study utilized breast cancer tissue samples and clinicopathological data obtained from 204 participants at the Department of Pathology, Second Affiliated Hospital of Zhejiang University Medical College, China. The dataset was randomly divided into two cohorts: 70% of the samples (142 cases) constituted the training set, while the remaining 30% (62 cases) formed the test set. To ensure the credibility and reliability of our research findings, we assessed the baseline characteristics of the clinical research subjects using the Ki-67 score and age. Upon conducting statistical analyses, we observed no significant differences between the two datasets, affirming the comparability of the training and test sets and underscoring the reliability of our subsequent analyses (Table 2). In the training set, 105 out of 142 cases included metastasis information, and these cases were utilized to establish metastasis prediction models. The test set, consisting of 48 out of 62 cases with available metastasis information, along with the entire dataset comprising 153 out of 204 cases with metastasis information, was employed to assess the performance of the previously established metastasis prediction models (Fig. 2). Similarly, 94 cases with survival information from the 142 cases in the training set were used to train the survival model, and the survival model was subsequently tested on the 47 cases with survival information in the test set. The flowchart illustrating the selection of patients and the process of dataset construction is presented in Fig. 2.

Table 2 The baseline characteristics of the participants.

Feature name	Train	Test	P-value	
Ki67	0.29 ± 0.24	0.27 ± 0.20	0.955	
Age	52.88 ± 11.00	52.32 ± 9.71	0.862	

Figure 2 The flowchart of patients selection. A total of 204 samples of breast cancer patients were included as training set (142 cases) and test set (62 cases).

Tumor area identification with convolutional neural networks (pathology analysis)

Patch level efficiency

In this study, we employed three convolutional neural network (CNN) models: DenseNet121, ResNet50, and Inception_v3, to predict tumor areas. These deep learning architectures are commonly used for various computer vision tasks, including image classification (Wang et al., 2024; Fatima, Rizvi & Rizvi, 2024). To assess the accuracy of these models in identifying tumor regions, we utilized patch-level ROC curves to compare and characterize the models. Furthermore, we visualized the aggregation of patches into WSI to evaluate the models’ performance. Predicted labels and probability heatmaps were obtained to facilitate the evaluation process. Table 3 and Fig. 3 display the patch-level accuracy and AUC of each model. Our results indicate that all three models achieved an AUC of over 0.95 in the training set, whereas AUCs in the test set were slightly lower, with 0.806 for Inception_v3, 0.800 for DenseNet121, and 0.790 for ResNet50, respectively. The results indicate that, for ER identification, the Inception V3 model exhibits particularly robust patch recognition capabilities. Consequently, both our PLH and BoW pipelines incorporate multiple instance learning feature aggregation based on the recognition outcomes of Inception_v3. Additionally, we conducted ER status prediction using the training set and test set with these three models based on supervised algorithms. Once again, the Inception_v3 model demonstrated robust patch recognition capabilities, achieving an AUC of 0.824 in the test set (Fig. S1).

Table 3 Displays the patch-level accuracy and AUC scores of each model.

Model name	Acc	AUC	95% CI	Sensitivity	Specificity	PPV	NPV	Cohort	
DenseNet121	0.903	0.967	[0.9665–0.9673]	0.854	0.931	0.881	0.915	Train	
DenseNet121	0.753	0.800	[0.7977–0.8014]	0.626	0.810	0.596	0.829	Test	
Inception_v3	0.888	0.958	[0.9571–0.9581]	0.831	0.922	0.863	0.902	Train	
Inception_v3	0.765	0.806	[0.8045–0.8082]	0.591	0.842	0.626	0.822	Test	
ResNet50	0.905	0.969	[0.9684–0.9692]	0.858	0.933	0.883	0.917	Train	
ResNet50	0.759	0.790	[0.7885–0.7923]	0.477	0.885	0.649	0.791	Test	
Note:

densenet121, Densely Connected Convolutional Networks 121; ResNet50, Residual Network 50.

Figure 3 The ROC curves for each model’s performance on the three datasets, namely Densenet121, ResNet50, and Inception_v3, arranged from left to right.

Visualize of predictions

For the visualization and potential interpretation of our tumor area prediction models, we applied the gradient-weighted class activation mapping (Grad-CAM) method to visualize the important regions reflecting the results of our CNN models. This visualization is achieved by examining the activation of the last convolutional layer of the model. Grad-CAM provides heatmaps that highlight regions in the input image where the model focuses its attention, helping to interpret and understand the decision-making process of the model. Accordingly, we assessed the spatial distribution of predictions made by the model at the per-slide level, aggregating the per-tile level prediction scores and mapping them back to the original slide dimensions in the form of heatmaps (Fig. 4). Within each slide, we observed that when tile-level CAM was aggregated at the per-slide level, the color scale on the right side of the corresponding image, from blue to red, the greater the weight of distinguishing tumor areas from non-tumor areas. The prediction result pertains to a total of 649,531 patches. It is evident that our pathological model exhibits a high level of accuracy in predicting tumor region tiles (Fig. 5).

Figure 4 Visualization of five patient examples.

Each example shows the tile image and corresponding heat map, and the red region represents a larger weight, which can be decoded by the color bar on the right (14-7569B3,14-7569B3,14-7800B3,14-22297A4,14-49849A1,14-8877A1).

Figure 5 Prediction and probability map of 13-32843A3,14-39673A4,13-5701A3 sample.

Metastasis analysis

To establish superior metastasis models, we applied random forest, extreme gradient boosting (XGBoost), and the light gradient boosted machine (LightGBM) methods to a training set of 105 samples on both supervised and weakly supervised basis to establish metastasis prediction models. These three methods have been applied in other literature and achieved notable results (Wang et al., 2024; Tang et al., 2023). We conducted predictions on a cohort of 153 participants with available metastasis information, applying features aggregated through multi-instance learning to the task of metastasis prediction. Through, multi-instance learning, we aggregated a total of 206 features, employing two distinct procedures. Each procedure yielded 101 probability features and two predictive label features. We subjected these features to correlation-based selection, retaining one of any pair of features with a Pearson correlation coefficient exceeding 0.9. This process resulted in a final set of 71 features, which we used for machine learning algorithm modeling. The performance metrics of the model are outlined below (Table 4). From our metastasis learning tasks, it was observed that under identical data conditions, while patch-level prediction AUC was higher for models based on supervised learning (Table 4 and Table S1), their performance in metastasis prediction was lower compared to models using weakly supervised learning (Fig. 6 and Table S2 and Fig. S2).

Table 4 Metrics in train and test cohort in predicting risk of metastasis.

Model name	Accuracy	AUC	95% CI	Sensitivity	Specificity	PPV	NPV	Task	
RandomForest	0.819	0.921	[0.8265–1.0000]	1.0	0.050	0.817	1.0	Train	
RandomForest	0.938	0.822	[0.5696–1.0000]	1.0	0.000	0.937	0.0	Test	
XGBoost	0.867	0.981	[0.9580–1.0000]	1.0	0.300	0.859	1.0	Train	
XGBoost	0.958	0.804	[0.5757–1.0000]	1.0	0.333	0.957	1.0	Test	
LightGBM	0.848	0.914	[0.8421–0.9868]	1.0	0.200	0.842	1.0	Train	
LightGBM	0.938	0.807	[0.4682–1.0000]	1.0	0.000	0.937	0.0	Test	

Figure 6 ROC in train cohort in predicting risk of metastasis.

ROC in test cohort in predicting risk of metastasis.

Survival analysis

Subsequently, we developed a survival prediction model using both supervised and weak supervision on a dataset of 94 samples with survival information in the training set. In survival analysis, models employing supervised learning algorithms, while demonstrating better accuracy in the training set than those based on weakly supervised learning (Fig. S3), did not exhibit a distinct advantage in the same test set. We evaluated the model results using the C-index, which yielded values of 0.892 and 0.710 in the training and testing sets, respectively. Figure 7A exhibits the results of KM analysis for the training cohort, while Fig. 7B showcases the KM analysis outcomes for the test cohort.

Figure 7 (Left) Kaplan-Meier (KM) analysis for the training cohort; (right).

Kaplan-Meier (KM) analysis for the test cohort.

Additionally, to more intuitively assess the patient’s condition and predict the probability of the patient’s clinical outcome, we constructed a Nomogram to assist clinicians in potentially making individualized treatment decisions. The nomogram, employing the Cox algorithm, was used to integrate clinical features and the pathomics signature. Figure 8 displays the nomogram for clinical application. Patients’ age, Ki-67 proliferation index, and the occurrence of metastasis are negatively correlated with survival time.

Figure 8 The nomogram for clinical application.

Discussion

The integration of deep learning methodologies in medical image analysis, particularly in the context of breast cancer pathology, has shown promising results in predicting metastasis and prognosis. Our study leveraged deep convolutional neural networks, including DenseNet121, ResNet50, and Inception_v3, to accurately identify tumor areas and predict ER status with high AUC values. Notably, our approach of utilizing patch-level prediction and multi-instance learning-based WSI feature fusion enabled robust performance in metastasis analysis. We observed that weakly supervised learning models outperformed supervised ones in predicting metastasis, underscoring the efficacy of this approach in capturing subtle patterns indicative of disease progression. Furthermore, our survival analysis models exhibited a C-index of 0.892 in the training set and 0.710 in the testing set, demonstrating their potential for stratifying patients into clinically significant groups. The development of a nomogram integrating clinical features and the pathomics signature further enhances the utility of our findings for personalized treatment decision-making in breast cancer management.

Histopathology serves as the cornerstone for cancer diagnosis, offering crucial information that guides treatment, management, and prognostication (Khened et al., 2021). Tissue pathology sections offer insights into various parameters such as tumor size, differentiation degree, histologic grade, cellular atypia, mitotic figures, and invasion characteristics, all of which are closely linked to treatment strategies, metastasis potential, and survival outcomes (Khened et al., 2021). While formalin-fixed paraffin-embedded sections are standard in histopathology, offering high-quality images albeit with prolonged preparation times, frozen sections serve as rapid solutions in emergency situations, despite being more susceptible to artifacts and morphological distortions (Cifci, Foersch & Kather, 2022; Kang et al., 2022; Sun et al., 2020). Despite the increasing emphasis on predicting tumor aggressiveness through genetics and omics data, histopathology remains pivotal for prognostic stratification (Kang et al., 2022; Shi et al., 2021). Recent studies have highlighted the potential of deep learning architectures in predicting patient outcomes from digital pathology images, with applications ranging from hepatocellular carcinoma prognosis to quantifying tumor-stroma ratio in colorectal cancer as an independent predictor of overall survival (Kang et al., 2022; Shi et al., 2021; Xu et al., 2023). Here, the utilization of deep learning techniques in our study has demonstrated promising results in predicting breast cancer metastasis and prognosis by effectively analyzing H&E stained WSI from radical breast cancer specimens.

Recently, computational tissue pathology features have demonstrated a strong prognostic correlation, significantly enhancing prognostic accuracy beyond traditional grading and histopathological typing alone. In breast cancer pathology, IHC staining plays a pivotal role in treatment decisions and outcome prediction, particularly in determining molecular subtypes based on biomarkers such as ER, HER2, and PR (Cifci, Foersch & Kather, 2022). Notably, Cifci, Foersch & Kather (2022), Shamai et al. (2019) developed a DL-based method capable of predicting 19 relevant biomarkers, including ER, PR, and HER-2, with precision comparable to IHC. Similarly, Cifci, Foersch & Kather (2022), Couture et al. (2018) reported an accuracy of 0.84 in predicting estrogen receptor status within their cohort analysis. Our study aligns with these findings, estimating the accuracy of ER status prediction to be approximately 0.84 within our specific research context. These advancements underscore the potential of deep learning methodologies in revolutionizing the assessment of biomarkers and molecular subtypes in breast cancer pathology, offering enhanced predictive capabilities for patient outcomes and treatment strategies.

However, our study has some limitations. Firstly, being a single-center study with limited data, the generalizability of our predictive results could be enhanced by future research incorporating multi-center data and expanding the dataset size. Secondly, our reliance solely on pathology genomics for model establishment suggests a potential avenue for improvement by integrating additional medical images such as radiology and ultrasound scans to enhance the model’s predictive capabilities. Furthermore, the retrospective nature of our study underscores the need for prospective clinical data from diverse clinical trials to bolster the robustness and applicability of our prediction model. Despite these limitations, our constructed prediction model demonstrates promising accuracy in forecasting patient metastasis and survival outcomes, offering significant clinical utility in treatment planning and outcome optimization.

The pathological model established by deep learning shows high accuracy in identifying tumor area and ER status, and has great potential in predicting metastasis and prognosis of breast cancer. Our research shows that the weak supervised learning model is superior to the supervised learning model in predicting transfer. In addition, nomograms integrating clinical and pathological features also provide a valuable tool for clinicians to make individualized treatment decisions for breast cancer.

Conclusion

In conclusion, our study has successfully developed a model utilizing weakly supervised deep learning on pathological H&E sections from 204 radical mastectomy cases to predict metastasis and survival outcomes in breast cancer. This innovative approach not only offers new avenues for tailoring treatment plans for breast cancer patients but also presents novel strategies for enhancing survival rates and prognostic accuracy. Notably, our findings highlight the superior performance of weakly supervised learning over supervised learning, emphasizing the potential for automated tumor area identification in future machine learning applications within breast cancer pathology. The implications of our research extend beyond predictive modeling, paving the way for advancements in personalized medicine and improved clinical outcomes in breast cancer management.

Supplemental Information

Supplemental Information 1 Classification and grouping of breast cancer.

Label 1: luminal breast cancer (ER positive), label 0: non-luminal breast cancer (ER negative); 70% of the total number as the training group and 30% of the total number as the testing group randomly.

Supplemental Information 2 Metastasis information of breast cancer.

Label 1: metastasis breast cancer, label 0: non-metastasis breast cancer.

Supplemental Information 3 Survival information of breast cancer.

Label 1: dead breast cancer patients, label 0: living breast cancer patients and survival time.

Supplemental Information 4 Supplementary material.

Additional Information and Declarations

Competing Interests

Author Contributions

Human Ethics

Data Availability

The authors declare that they have no competing interests.

Hui Liu conceived and designed the experiments, analyzed the data, prepared figures and/or tables, authored or reviewed drafts of the article, and approved the final draft.

Linlin Ying performed the experiments, prepared figures and/or tables, and approved the final draft.

Xing Song performed the experiments, prepared figures and/or tables, and approved the final draft.

Xueping Xiang performed the experiments, prepared figures and/or tables, and approved the final draft.

Shumei Wei analyzed the data, authored or reviewed drafts of the article, and approved the final draft.

The following information was supplied relating to ethical approvals (i.e., approving body and any reference numbers):

This study was approved by the Human Research Ethics Committee of the Second Affiliated Hospital of Zhejiang University School of Medicine (approval No.2023 Lun Shen Yan No. (1035)). Surviving patients know and provide informed consent.

The following information was supplied regarding data availability:

The raw measurements are available in the Supplemental Files.

The machine learning data that support the findings of this study are available at GitHub:

- https://github.com/OnekeyAI-Platform

The code are available at GitHub and Zenodo:

- https://github.com/OnekeyAI-Platform/ML-of-BC.

- Liu, H. (2024). Development of metastasis and survival prediction model of luminal and non-luminal breast cancer with weakly supervised learning based on pathomics [Data set]. Zenodo. https://doi.org/10.5281/zenodo.14536316.

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
