# Peer review of "Development of metastasis and survival prediction model of luminal and non-luminal breast cancer with weakly supervised learning based on pathomics"

_PeerJ, doi:10.7717/peerj.18780_

## Round 0.1 · original submission · Major Revisions

1. The manuscript contains multiple unclear paragraphs and sentences. A thorough language edit is necessary to improve readability and ensure the intended meaning is conveyed accurately. For example, terms like "transfer" (Line 230-238) are used ambiguously. Please carefully construct each paragraph with 3-8 sentences for better flow and comprehension.
2. More details are needed on the three neural network models (DenseNet21, ResNet50, Inception_v3) and the machine learning models (RandomForest, XGBoost, LightGBM). Clarify how they were implemented, compared, and evaluated.
3. Please provide specifics on the training and testing data split, sampling methods, and evaluation metrics used. The current focus on AUC in a small testing group is insufficient to support the conclusions.
4. Expand the introduction with a more comprehensive review of related work on predicting metastasis and survival in breast cancer. Highlight how your study differs from and builds upon existing research.
5. Clarify the specific aim and novelty of your work - is the focus on predicting metastasis/survival, or classifying luminal/non-luminal subtypes? Ensure this is consistent throughout.
6. Resolve the discrepancy between mentioning three datasets in the discussion versus stating data was from a single hospital in the methods.
7. Compare your models' performance to existing state-of-the-art methods to better contextualize the findings.
8. Discuss the practical implications and potential clinical applications of the work.
9. Use proper statistical terms (e.g., Shapiro-Wilk test) and specify the type of t-tests employed.
10. Add citations for software packages used.

Reviewer 1 ·

Basic reporting

There are multiple inexplicit paragraphs throughout the paper, please carefully revise the english sentences and words before considering a resubmission. For example, from Line 230 to Line 238, samples with transfer information, in here "transfer" have inexplicit meanings, referring to tumor samples with metastasis information or transfer learning model?

Experimental design

The biggest issue of the paper is the methodology and result part, which are not well written and messed up. The author mentioned three neural network models: DenseNet21, ResNet50, and Inception_v3. Also, mentioning the RandomForest, XGBoost, and LightGBM models. But all of them are lack of details, and the comparison results are not clear. The evaluation metrics of the models are not clear, by only considering AUC in the small testing group. How the training and testing data samples are split and sampled are ambiguous. By only stating the AUC in the small testing group is too weak to support the conclusion of the paper. Would like to see more validation on external datasets.

Validity of the findings

The scientific meanings and findings in the paper described is not clear. The paper messed up for delivering a predictive model for predicting metastasis or survival of breast cancer, or trying to predict luminal or non-luminal breast cancer. But throwing some machine learning models and doing some kind of predictions.

Reviewer 2 ·

Basic reporting

1. The authors have made a commendable effort, and I consider the manuscript a valuable contribution. However, the study lacks support from a significant and relevant background review. Much similar work has recently been done on predicting metastasis and survival based on breast cancer. It is therefore recommended to include a more comprehensive literature review to highlight the work already accomplished in this field.

2. Building on the previous point, it is also suggested that the authors provide evidence of how this work differs from prior studies.

Experimental design

1-The discussion section mentions three datasets, but the methods and materials section states that the data was collected from the Hospital of Zhejiang University Medical College. Could the author have mistakenly referred to models as datasets? Please clarify this discrepancy.

Validity of the findings

1- The authors are encouraged to compare the results of their proposed models with those of existing state-of-the-art models.

Additional comments

NA

·

Basic reporting

See my comments.

Experimental design

See my comments

Validity of the findings

See my commments

Additional comments

Thanks for opportunity to review manuscript entitled ‘‘Development of metastasis and survival prediction model of luminal and non-luminal breast cancer with weakly supervised learning based on pathomics’’ for Peerj Journal. The authors examined efficacy of models for predicting luminal and non-luminal breast cancer using three-networks for machine learning models. The strength of the manuscript include examining an important topic that cause significant human and public burden. Unfortunate, the article contains several important weaknesses. Because my main philosophy of reviewing a manuscript as reviewer and sometimes an editor to improve the manuscript and not punishing the authors, I provided very specific and detailed peer review of the manuscript to increase its quality and citation potential. I hope authors of the manuscript may benefit from my review. Necessary revisions reported section by section with the page and line number and when possible with suggestions.
Necessary Revisions
General
1. Authors must provide long name of abbreviation in its first use such as ER, DenseNet121, ResNet50, and Inception_v3 AUC
Abstract
2. A sentence must not be begun with but following sentence must be corrected ‘ ‘But the AUC of the metastasis prediction in training and testing sets is higher than value based on supervised learning.’’.
3. If authors used the C-index of the survival prediction model, they must give information about it in Method in abstract.
4. Practical implications of study findings in abstract is completely missing and must be added to Conclusion section as a last sentence.
Introduction
5. Introduction: must correct as Introduction
6. The citation/citations needed for following sentence ‘‘Breast cancer is the most common type of malignancy among women worldwide, with a high incidence and mortality rate.’’
7. The citation/citations needed for following sentence ‘‘Breast cancer is a biologically heterogeneous disease, with several recognized molecular subtypes that correspond to different responses to treatments and prognoses.’’
8. The citation/citations needed for following sentence ‘‘In the 2011 St. Gallen International Breast Cancer Conference, most experts reached a consensus on the classification of breast cancer based on immunohistochemical detection of estrogen receptor (ER), progesterone receptor (PR), human epidermal growth factor 2 (HER2), and Ki-67, which resulted in the classification of breast cancer into four molecular subtypes: luminal A, luminal B, HER2 overexpressed, and basal-like’’
9. The citation/citations needed for following sentence ‘‘In addition, basal-like subtype commonly metastasizes to the brain, bones, lungs, and liver’’
10. The citation/citations needed for following sentence ‘‘Therefore, current guidelines recommend neoadjuvant therapy for ER negative subtype breast cancers.’’
11. The citation/citations needed for following sentence ‘ ‘Histopathology tissue analysis is considered the gold standard in cancer diagnosis and prognosis.’’
12. The citation/citations needed for following sentence ‘ ‘Histopathology tissue analysis is considered the gold standard in cancer diagnosis and prognosis. Immunohistochemical (IHC) staining of breast cancer tissues has become widely used for breast cancer subtyping.’’
12. The citation/citations needed for following sentence ‘‘In addition, IHC92 based ER levels may be affected by poor quality samples.’’
13. Introduction, General: Introduction section is very weak for a scientific article. Firstly, authors must give information about previous studies and their weakness in Introduction section.
14. Introduction, General: Second, authors must give information about importance of supervised learning modeling based on pathomics on identifying luminal and non-luminal breast cancer.
Method
15. Are patients give consent to use their information? If so, authors must give information about it Patients and follow-up section.
16. Patients and follow-up must rename as Participants.
17. Authors must give more information about Pathology Procedures.
18. In the following ‘ ‘We utilized the Shapiro method to conduct statistical tests for assessing the normality of 218 clinical features. Subsequently, we employed the t-test to analyze the significance of these clinical features.’’ Shapiro method must be Shapiro-Wilk test. Moreover in the subsequent sentence t-tests is unclear dependent sample t-test or independent samples-t-tests.
18. In the following ‘ ‘Python code written in Python version 3.7.12. The Python packages used in the analysis encompassed Pandas version 1.2.4, NumPy version 1.20.2, PyTorch version 1.8.0, Onekey version 3.1.3, OpenSlide version 1.2.0, SciPy version 1.7.3, scikit-learn version 1.0.2, and Lifelines version 0.27.0.’2 authors must provide citations for used packages.

Results
19. Survival Analysis must be bold.
20. In all tables authors must clearly indicate training samples. Possible first represent training samples but it is unclear in tables.
21. All tables must be corrected as per APA-7 or AMA rules.
Discussion
22. Authors must add a introduction sentence to Discussion such as this study examined…….
23. Practical implications of study findings are are completely missing in Discussion section and must be added.
Conclusion
24. No severe problem exists in this section.
Manuscript General
25. Authors must construct each paragraph at least three at most eight sentences along the manuscript.
26. Extensive English editing required for this article.

---

## Round 0.2 · Minor Revisions

Authors have made full revisions according to reviewers' and editor's comments. Please address reviewer 3's comments.

·

Basic reporting

See my comments.

Experimental design

See my comments.

Validity of the findings

See my comments.

Additional comments

Thanks for opportunity to review revised manuscript entitled ''Development of Metastasis and Survival Prediction Model of luminal and non-luminal Breast Cancer with weakly supervised learning based on Pathomics'' for Peerj journal. Authors significantly improved in this revision to article. I congratulate them. However, as a experienced reviewer and editor, I think some revisions still required.
1. Abstract: must correct as Abstract
2. Introduction: must correct as Introduction.
3. begin a new paragprah in the following Breast cancer is generally classified into luminal and non-luminal types based on ER status. Compared to luminal types, non-luminal breast cancers tend to be more aggressive and are associated with a poorer prognosis[17]. Studies have been conducted based on deep learning to predict the classification of luminal and non109 luminal breast cancer through radiomics and ultrasoundomics [18]. However, there are a scarcity of models predicting the metastasis and prognosis of breast cancer with different molecular typing using machine learning through pathological sections of radical specimens. In this study, our aim was to develop and validate a deep learning pathomic model to predict the metastasis and prognosis of breast cancer
4. Before begining new paragraph to above add advanteges of using your supervised learning techniques to prediction Model of luminal and non-luminal Breast Cancer at least one prefrablt two paragraph.
5. [# 2023 Lun Shen Yan No. (1035)]. I am not able to understand this?
6. Add at least one pragraph to practical implicatinos of your findings after limitations and before conclusions, only one sentence is not enough.
7. dont use dashed lines in table 1. (black-white font, all white)
8. dont use dashed lines in table 2.
9. dont use dashed lines in table 3. Ad a note section and write long names of densenet121, densenet121, resnet50 Note. densenet121 = ........
10. the above comment also valid for table 4.

---

## Round 0.3 · accepted · Accept

Authors have fully improved this manuscript, and I think this manuscript can be accepted for publication.

·

Basic reporting

See my comments.

Experimental design

See my comments.

Validity of the findings

See my comments.

Additional comments

Thanks for opportunity review revised manuscript entitled ‘‘Development of Metastasis and Survival Prediction Model of luminal and non-luminal Breast Cancer with weakly supervised learning based on Pathomics’’. I would like the thanks to authors. They make a good job for improving quality of their manuscript. Authors revised the manuscript as I requested with a good will. In this form, Introduction reflects very well the previous studies and study aim, Method section and Result section is correct, and Discussion section adequately synthesis to previous study findings and current study results. Overall, I have no further comment regarding to manuscript. I congratulate to authors and wish them success on their future endeavors.